# Multi-Objective Optimization Design of Vehicle Side Crashworthiness Based on Machine Learning Point-Adding Method

**Dawei Gao *** , **Bufan Yao** , **Gaoshuang Chang and Qiang Li**

School of Mechanical Engineering, University of Shanghai for Science and Technology, Shanghai 200093, China
* Correspondence: gddwww1999@163.com

**Abstract:** Multi-objective optimization problems are often accompanied by complex black-box functions which not only increases the difficulty of solving, but also increases the solving time. In order to reduce the computational cost of solving such multi-objective problems, this paper proposes an ARBF-MLPA (Adaptive Radial Basis Function neural network combined with Machine Learning Point Adding) method, which uses an ABRF (Adaptive Radial Basis Function) neural network and OLHS (Optimized Latin Hypercube Sampling) to establish the first generation metamodel and uses the NSGA-II (Non-dominated Sorting Genetic Algorithm II) optimization algorithm to obtain the optimal front edge of Pareto. The ARBF-MLPA method is continuously used to select and add points to update the meta-model, then dynamically improve the accuracy of the meta-model until the optimal front edge converges. Then the ARBF-MLPA method and RBF-UDPA (Radial Basis Function neural network combined with Uniform Point Adding) method are compared using the test functions of three different frontier features. The performance evaluation indexes of Inverted Generation Distance (IGD), Hypervolume (HV) and Spacing Metric are superior to RBF-UDPA. Finally, ARBF-MLPA method combined with the NSGA-II optimization algorithm is applied in the multi-objective optimization design of vehicle-side crashworthiness. The model converges after 6 iterations. Comparing the results obtained by the ARBF-MLPA method with the finite element simulation results, the error is within 5%, which meets the error requirements. The optimized model reduces chest intrusion by 4.32%, peak collision force by 2.11% and reduces mass by 14.05%.

**Keywords:** multi-objective optimization design; machine learning point-adding method; adaptive radial basis function neural network; vehicle-side crashworthiness

## 1. Introduction

The continuous increase in car usage ensures environmental damage and road accidents, among all traffic accidents, vehicle accidents contributed the highest injury and death toll. Vehicle driving safety is divided into active safety and passive safety. As a vehicle passive safety accident, vehicle-side collision causes more injury and death than any other accident of this type. For this reason, this paper chooses to study the safety of vehicle-side impact and aims to reduce the mass of the whole vehicle on the premise of ensuring passenger safety in order to achieve the objectives of energy conservancy, emission reduction and improving passive safety of vehicles.

Vehicle crashworthiness is a kind of multi-objective topic in engineering, vehicle-side impact simulation involves highly nonlinear computation, and the optimization process involves multiple iterations. Therefore, the optimization analysis, merely using finite element simulation, is very inefficient. In order to improve the optimization process, now a variety of metamodel methods, e.g., Response Surface Method (RSM) [1], Kriging Model [2], Artificial Neural Network (ANN) [3], Support Vector Machine (SVM) [4] and Radial Basis Function (RBF) [5] and Polynomial Method [6] have been developed. However, in solving engineering problems, it takes plenty of time to establish high-precision metamodels, and

the local optimal solution is a common dilemma in solving highly nonlinear engineering problem [7]. For this reason, it is already very difficult for the conventional mathematical surrogate models to satisfy engineering requirements. To address this problem, automobile engineers have proposed the method of using a mathematical surrogate model combined with sequential point adding [8,9]. Additionally, they improved the accuracy of the model by continuously updating surrogate models, so as to avoid falling into the dilemma of local optimal solution and gradually approach optimal solutions.

The sequential point-adding strategy optimization algorithm of these years is developing rapidly. Among all scholars, Chan Jing of Jilin University [10] discovered a Multi-Objective Particle Swarm Optimization algorithm (KMOPSO) based on the Kriging model point-adding strategy; I. Y. Kim et al. [11] proposed a double-objective optimization method having multi-objective application potential, an adaptive weighted-sum method changing the previous weighted-sum approach; Jianhua Zhou et al. [12] invented a new high-efficiency sequential multi-objective optimization method (S-MOO), in which the anchors in the design space of global variables are fully leveraged to generate a dataset of global solutions to guide the search for Pareto solutions. These sequential point-adding strategies have evident strength in solving high latitude, nonlinear multi-objective problems. Not just having fast convergence speed, they can also quickly approach Pareto front edge.

Combining sequential point-adding optimization algorithm and engineering problems can increase the optimization speed of engineering problems and save computation cost significantly. As below are some applications of vehicle crashworthiness.

Sun et al. [13,14] put forworda parallel Multi-Objective Efficient Global Optimization (MOEGO) algorithm, which utilizes several local optimal strategies of "Kriging Believe" and EI strategies to realize parallel computation. It uses HV (hypervolume) as the indicator of beam multi-objective convergence, addresses the design of crashworthiness and tests it in practical applications, and further proves the effectiveness and accuracy of the algorithm. Fang Jianguang et al. [15] raised a new design scheme for custom welded structures of mixed materials, which applies sequential sampling strategy to establish surrogate model, realizes lightweight design on the premise of improving the rigidity of car door structure. Hangfeng Yin et al. [16] invented an adaptive radial set function method based on metamodel, which solves MOO problem well, and they applied this method to the crashworthiness optimization for a new thin-wall structure.

To sum up, sequential optimization strategy has walked from theoretical level to the applications in engineering optimization problem, and it is of very good application importance in multi-objective optimization design of vehicle crashworthiness. For this reason, this paper proposed an Adaptive Radial Basis Function combined with a Machine Learning Point-Adding (ARBF-MLPA) method.

In this paper, an ARBF neural network combined with MLPA is proposed. The performance of the proposed algorithm is tested by using mathematical test function, and it is applied to the engineering problem of vehicle-side crashworthiness optimization.

1. Using ABRF neural network and OLHS to establish the first generation metamodel, the NSGA-II optimization algorithm is then used to obtain the Pareto, and the MLPA is combined to dynamically improve the accuracy of this surrogate model until it converges.
2. Different from the traditional method of adding points, this paper mainly considers the maximum–min distance (MMD) point, the nearest boundary (NB) point and the high-quality points on the optimal front edge obtained by ARBF-MLPA. The selection of these three points helps to improve the local accuracy of the model and leads the optimization results being closer to the real front edge.
3. After test functions of three different frontier features testing, the method proposed in this paper has more advantages than the traditional RBF-UDPA method in the overall convergence speed and the distribution of the final optimal front.

4. The head, chest, abdomen and pelvis of the driver were corresponding to the corresponding position of the B-pillar of the car, and the amount of invasion of each part was measured indirectly by setting the spring unit at the B-pillar, so as to simulate the invasion injury suffered in the side impact of the car. The amount of chest invasion was included as one of the design variables.

5. Combine the ARBF-MLPA method with the NSGA-II optimization algorithm to optimize the design for vehicle-side crash safety The optimized model reduces chest intrusion by 4.32%, peak collision force by 2.11% and reduces mass by 14.05%.

The method proposed in this paper with higher accuracy and converges faster than the traditional point-adding method. It can be used in the multi-objective optimization design of the side crashworthiness of the vehicle, which can reduce the optimization process time on the premise of ensuring the correct calculation.

## 2. ARBF-MLPA Method

### 2.1. Design of Experiment (DOE)

In general, before a metamodel is established, design samples are usually generated in design space applying different design of experiment (DOE) methods such as Full factorial, Latin hypercube sampling (LHS), etc. Among all, LHS is widely applied due to its flexibility in data density and location, and optimal Latin hypercube sampling (OLHS) is a sampling method developed on the basis of LHS. Its samples selected are more uniform than LHS [17,18]. Accordingly, this paper applies an OLHS method to generate initial samples.

### 2.2. Radial Basis Function (RBF) Neural Network

In machine learning, neural network is a good model for forecast and analysis. There are several types of neural network. Among these, radial basis function neural network features optimal approximation, fast convergence speed, and the ability to overcome the problem of local minimal value. In addition, a radial basis network can approach any continuous function with any accuracy. Radial basis models mainly take the basis function as the sampling point symmetry center to build radial basis function model for discrete multivariate data interpolation. Radial basis function is usually expressed in expression (1):

$$\hat{y}(x) = \sum_{j=1}^{m} c_j p_j(x) + \sum_{i=1}^{n_s} \lambda_i \varphi(r(x, x_i)) \tag{1}$$

where $m$ represents the number of items in polynomial expression, $c_j$ represents the coefficient of polynomial basis function $p_j(x)$, $n_s$ represents the number of sampling points, $\lambda_i$ represents the weighting coefficient of variable $i$, $r(x, x_i)$ represents Euclidean distance, and $\varphi(r)$ represents radial basis function.

In radial basis function models, the selection of basis function is critical. There are the following types of basis functions shown in Table 1, in which $r$ represents Euclidean distance, and $c$ represents shape parameter, which can take a random value you desire.

**Table 1.** RBF common basis functions.

| Basis Function | Mathematical Expression |
|---|---|
| Linear | $\varphi(r) = cr$ |
| Cubic | $\varphi(r) = (c + r)^3$ |
| Thin plate | $\varphi(r) = r^2 \ln(cr^2)$ |
| Gaussian | $\varphi(r) = \exp(-cr^2)$ |
| Multi-quadric | $\varphi(r) = \sqrt{c^2 + r^2}$ |
| Inverse multi-quadric | $\varphi(r) = 1/\sqrt{c^2 + r^2}$ |

In view that vehicle crash is a complicated nonlinear function-changing process, the main advantage of RBF is that it can approximate nonlinear functions very well, can handle the laws that are difficult to analyze in the calculation process, and has good generalization capability and extremely fast convergence. For this reason, this paper selects RBF of Gaussian basis function to build the metamodel of whole vehicle-side crash optimization.

### 2.3. Multi-Objective Optimization Algorithm

Genetic Algorithm (GA) is a very popular global optimization algorithm originated from natural evolution mechanism and principles of genetics. GA is better than many traditional optimization algorithms, as it avoids the problem of local minimal value while being able to search for optimal values. Non-dominated Sorting Genetic Algorithm (NSGA) is very effective in solving multi-objective problem. It arranges sequences applying non-dominated sorting solutions, and distributes fitness based on the sequences arranged. As an algorithm improved from NSGA, NSGA-II has been proved an effective multi-objective optimization algorithm in a lot of benchmark problems [19,20].

NSGA-II is also a multi-objective optimization method discussed based on Pareto optimal front edge. The final objective of this paper is to find the optimal Pareto front edge of all multi-objective optimization problems, which exactly coincides with the aim of the paper to find the solution of optimization. In addition, what we are building is a double-objective optimization model, but NSGA-II algorithm is enough. See Figure 1 for flow chart of NSGA-II algorithm and specific optimization process is as follows:

1. Firstly, the sample points obtained by OLHS method, and the response values obtained by finite element method, were used to construct the RBF metamodel;
2. In this algorithm, the design point is the first-generation population. The population is initialized, and the offspring is obtained through crossover, selection, mutation;
3. Merge the parents with the offspring and sort them non-dominated;
4. After sorting according to the rank, individuals are added to the next population in order to form a new parent population, and the remaining individuals will be eliminated after the number of the next generation arrives. This is the elite retention strategy based on the calculation of crowding degree;
5. If no new parent population is generated, the parent population of this generation will be sorted in a non-dominated way and a new parent generation will be formed from them. If a new parent population is created, the process of 2 to 4 is repeated until the termination condition is reached. Here, we set the maximum number of evolutionary generations as the termination condition;
6. Output the Pareto front of this optimization;
7. Using the MLPA method, select points in the Pareto front output from the last step. Additionally, the finite element method should be used to calculate the corresponding response values, so as to update the RBF model above.

### 2.4. Adaptive Radial Basis Function Neural Network Combined with Machine Learning Point-Adding Algorithm (ARBF-MLPA)

This paper proposes an adaptive radial basis function neural network combined with a machine learning point-adding method (ARBF-MLPA). The main idea of this method is: establish a mathematical surrogate model through ARBF neural network, obtain the optimal front edge through multi-objective optimization algorithm described above, find the original design sample points included in the optimal front edge, and find the original design sample points contained in the optimal front edge, add new sample points uniformly between the original sample points, and obtain the corresponding response values through calculation, add the newly obtained sample points to the original model, rebuild a new mathematical surrogate model, then continually update the surrogate model until the model converges. Combining the ARBF neural network with the MLPA algorithm makes model converge faster and continuously approach the optimal solution set.

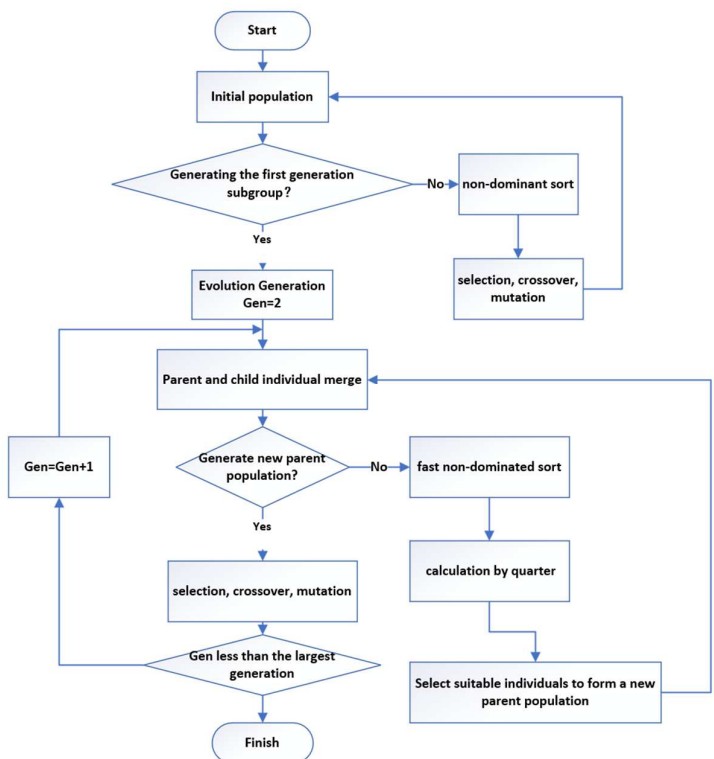

**Figure 1.** Flow chart of NSGA-II algorithm.

The point-adding method used in this paper mainly considers Max–Min Distance (MMD) point, Nearest Boundary (NB) point and the sample points selected by the ARBF neural network.

As below is the algorithm description in details.

Step 1: set initial parameters, e.g.: number, target, number of variable, and designed range of variable of initial samples.

Step 2: generate initial sample point based on the OLHS DOE described above.

Step 3: obtain the values of the objective function and constraint function at the initial sampling point. In practical engineering problems, the values of objective function and constraint function are usually obtained using finite element simulation calculation, so this step is a relatively time-consuming process.

Step 4: build a radial basis function neural network model based on the sample points obtained in step 3 and relevant response values.

Step 5: in the metamodel obtained in step 4, solve the multi-objective problem applying NSGA-II algorithm to obtain the Pareto optimal front edge, and normalize the obtained response value of Pareto front edge in iteration to improve the accuracy of the model.

Step 6: add the high-quality point on the optimal front edge obtained through ARBF-MLPA to the model to be built next time, as training sample, these points will lead the optimization results to be closer to the real front edge. As shown in Figure 2, the sample points added in iteration each time are uniformly distributed between neighboring design sample points which are farther from Pareto front edge. As shown in scenario 1 in Figure 2, there are two large gaps between the design points distributed in the Pareto front, and we can uniformly select points between the two gaps as sample points to add to the model. As shown in scenario 2 in Figure 2, there are four design points in the Pareto front, but the gap between the two points is large, so we can uniformly select points in the middle of the large gap as sample points. Different from the traditional uniform point-adding method, this method is purposeful and selective in a certain design sample gap, rather than blindly uniform adding points.

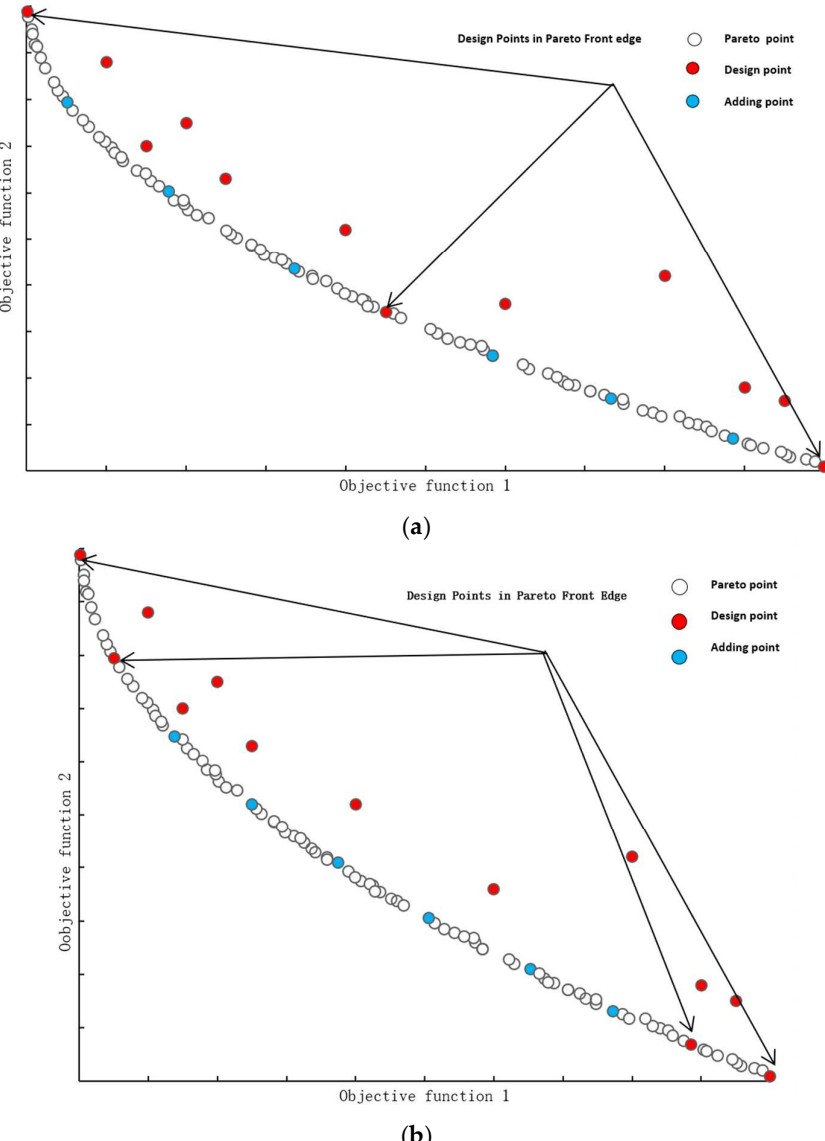

**Figure 2.** Pick extra sample points in Pareto points. (**a**) Scenario 1. (**b**) Scenario 2.

Step 7: select the extreme points in the Pareto optimal front edge corresponding to each target. The number of extreme points should be equal to the number of targets. This paper is presenting a two-target problem, and two extreme points should be selected.

Step 8: in the Pareto approximate optimal front edge, select the (Min) point set which is the nearest to the existing training sample, then select (Max) point which has greater relative distance from these point sets. As is generally the case, the model accuracy will be very low in sparse regions since MMD is exactly in such regions. So, these points are selected and taken as a new training sample, and this helps improve the local accuracy of models [21–23]. MMD points of Pareto front can be selected using the equation below:

$$\max\left[\min_{1\leq i\leq s,1\leq j\leq t}^{x_i\in S,x_j\in P}\left(d\left(x_i,x_j\right)\right)\right] \tag{2}$$

where, $x_i$ and $x_j$ respectively represent sample $i$ and sample $j$ of Pareto front point pool ($S$) and sample pool ($P$) in the last iteration; $d\left(x_i,x_j\right)$ represents Euclidean distance between $x_i$ and $x_j$.

Step 9: search for the Nearest Boundary (NB) points from each MMD point sample (obtained in step 7) and generate MMD-NB point, using the equation below:

$$x_{MMD-NB} = \left( \frac{x_{MMD} + x_{NB}^{EP}}{\alpha_1} \right)\beta_1 + \left( \frac{x_{MMD} + x_{NB}}{\alpha_2} \right)\beta_2 \tag{3}$$

wherein, $x_{NB}^{EP}$ represents the extreme point that satisfies the closest Euclidean distance to the design feasible region, and $x_{NB}$ represents the corresponding NB sample points; $x_{MMD}$ represents MMD sample point, $x_{MMD-NB}$ represents the obtained MMD-NB point; $\alpha_i$ represents the number of designed variables, which is 2 in this paper, $\beta_i$ represents threshold value, which is set to be 0.5 here, Figure 3 below shows the distribution of MMD-NB sample points in 2D space.

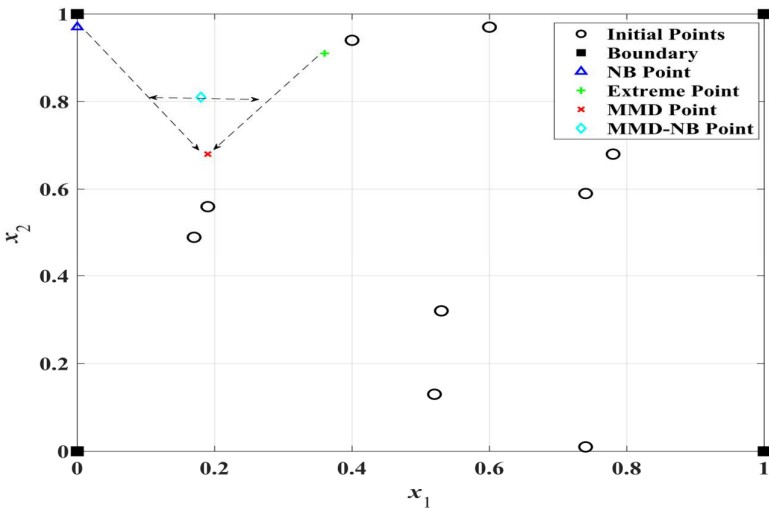

**Figure 3.** The distribution of MMD-NB in 2D space.

Step 10: add the response value of the new sample point obtained through FEM simulation to previous generation of sample pool, build a new mathematical surrogate model, and repeat the above operations till the computation results converge.

*2.5. Performance Test of Optimization Algorithm*

2.5.1. Test Function

To test the performance of the optimization algorithm described above, this paper tests the algorithm using the test functions of three optimal front edge shapes, i.e.: ZDT1, ZDT2, ZDT3 [24]. In the meantime, the traditional radial basis uniform point-adding method: Radial Basis Function combined with Uniform Distribution Point Adding (RBF-UDPA), which is similar to the point addition method in this paper, is used for comparison. The optimal front edges of these three test functions are convex, concave and discontinuous, and have certain characteristics which can well test the performance of the algorithm. Table 2 shows the range of variable, function expression and optimal front features of the three functions in details.

2.5.2. Performance Evaluation Indicators and Convergence Criteria

In order to judge the status of Pareto front edge convergence in the process of computation iteration, this paper rates the optimal front calculated using the ARBF-MLAP method described in this paper from different dimensions. The rating is made using the three evaluation indicators IGD (Inverted Generation Distance), HV (Hypervolume) and Spacing Metric, respectively.

**Table 2.** Test function.

| Function | Range of Variable | Function Expression | Features of Front |
|---|---|---|---|
| ZDT1 | [0, 1] | $\begin{cases} \min\ f_1(x_1) = x_1 \\ \min\ f_2(x_2) = g\left(1 - \sqrt{f_1/g}\right) \\ g(x) = 1 + 9 \sum\limits_{i=2}^{m} x_i/(m-1) \end{cases}$ | Continuous convex |
| ZFDT2 | [0, 1] | $\begin{cases} \min f_1(x_1) = x_1 \\ \min f_2(x) = g\left(1 - (f_1/g)^2\right) \\ g(x) = 1 + 9 \sum\limits_{i=2}^{m} x_i/(m-1) \end{cases}$ | Continuous concave |
| ZDT3 | [0, 1] | $\begin{cases} \min f_1(x_1) = x_1 \\ \min f_2(x) = g\left(1 - \sqrt{f_1/g} - (f_1/g)\sin(10\pi f_1)\right) \\ g(x) = 1 + 9 \sum\limits_{i=2}^{m} x_i/(m-1) \end{cases}$ | Discontinuous |

IGD: a comprehensive evaluation index for the convergence of multi-objective optimization algorithms. The smaller the value, the better the convergence of the algorithm and the distribution of the front edge [25,26].

HV: invented by Zitzler et al. [27,28]. It refers to the volume of the hypercube enclosed by the individuals in the solution set and the reference point in the target space. The bigger HV value is, the better the comprehensive performance of the algorithm is.

Spacing metric: an indicator proposed by Deb et al. in 2002, which is mainly used to judge the distribution degree of non-dominated solutions. The lower this value is, the better the performance of the algorithm is.

2.5.3. Test of Mathematical Functions of Optimization Algorithm

This section tests the convergence performance of ARBF-MLAP algorithm using the test functions and performance evaluation indicators described above. In the meantime, in order to verify the effectiveness and fast convergence of the ARBF-MLAP method, the RBF-UDPA method is selected for comparison. The traditional RBF point selection does not have the advantages of ARBF. For the sake of test fairness, 5 sample points are uniformly selected on the optimal front in finding the solution of the multi-objective function established by RBF, in order to be consistent with the point selection method of ARBF. So, this uniform point selection method is called the radial basis function uniform point-adding (RBF-UDPA) method. The selection of MMD and MMD-NB points uses the same method as that of RBF and ARBF.

For details of the iteration of different evaluation indicators of RBF-UDPA and ARBF-MLPA, function and the comparison between test front and real front edge are shown under each test in Figures 4–7.

As shown in Figures 4 and 5, under the two test functions (ZDT1 ZDT2), ARBF-MLPA converges at generation 8, while RBF-UDPA does not converge until generation 10. In addition, based on IGD, Spacing and HV evaluation criteria, ARBF-MLPA is superior to RBF-UDPF in all performance indicators.

As shown in Figure 6, under the test function ZDT3, ARBF-MLPA converges at generation 9, while RBF-UDPA does not converge until generation 13. In addition, based on IGD, Spacing and HV evaluation criteria, ARBF-MLPA is superior to RBF-UDPF in all performance indicators.

As shown in Figure 7, ARBF-MLPA is closer to the real front edge of three test functions.

To sum up, ARBF-MLPA is superior to RBF-UDPA in terms of total convergence speed and the final optimal front distribution. Accordingly, ARBF-MLPA can save computation time and obtain more accurate solution in solving multi-objective problems and in its application to practical engineering problems.

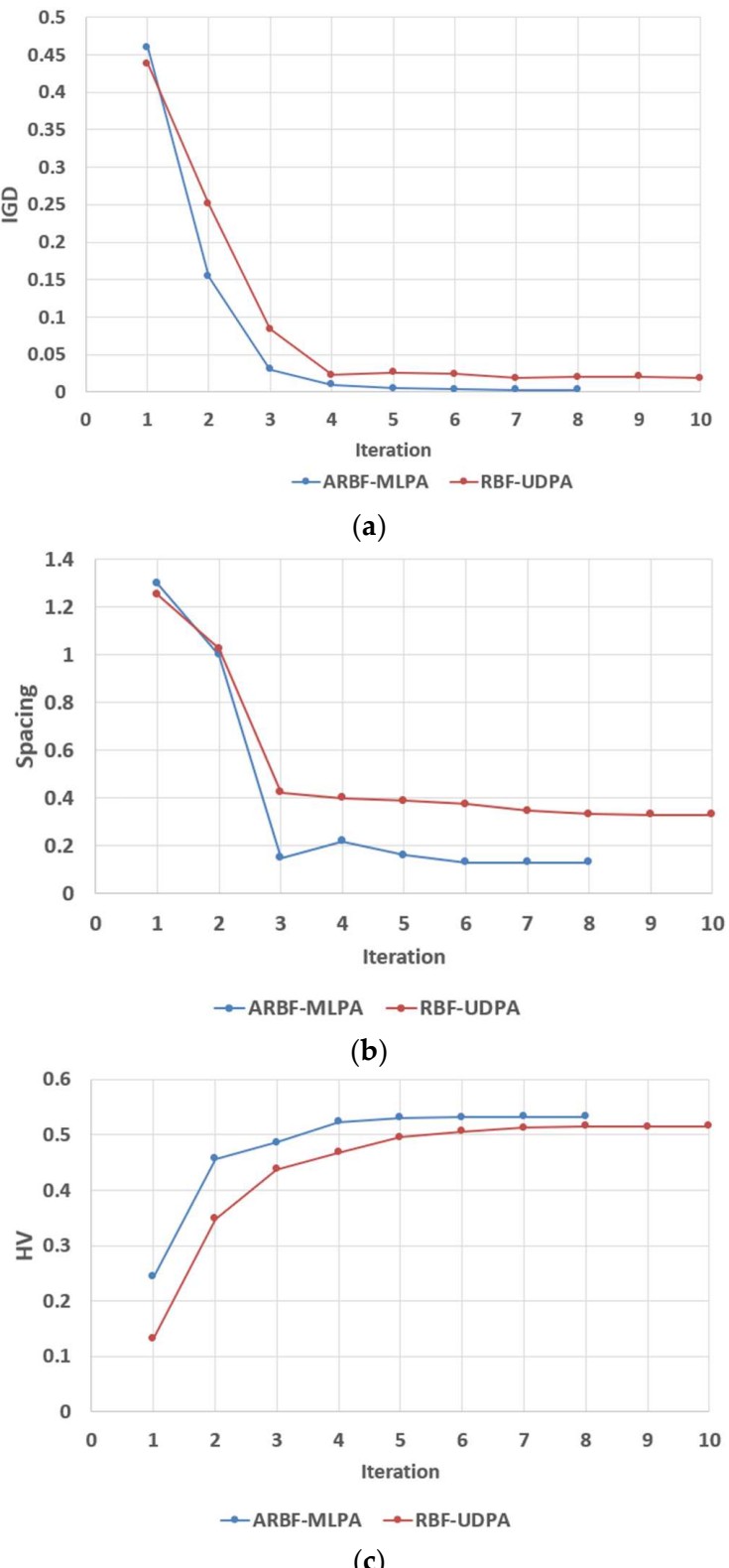

**Figure 4.** (**a**) IGD under function ZDT1. (**b**) Spacing under function ZDT1. (**c**) HV under function ZDT1.

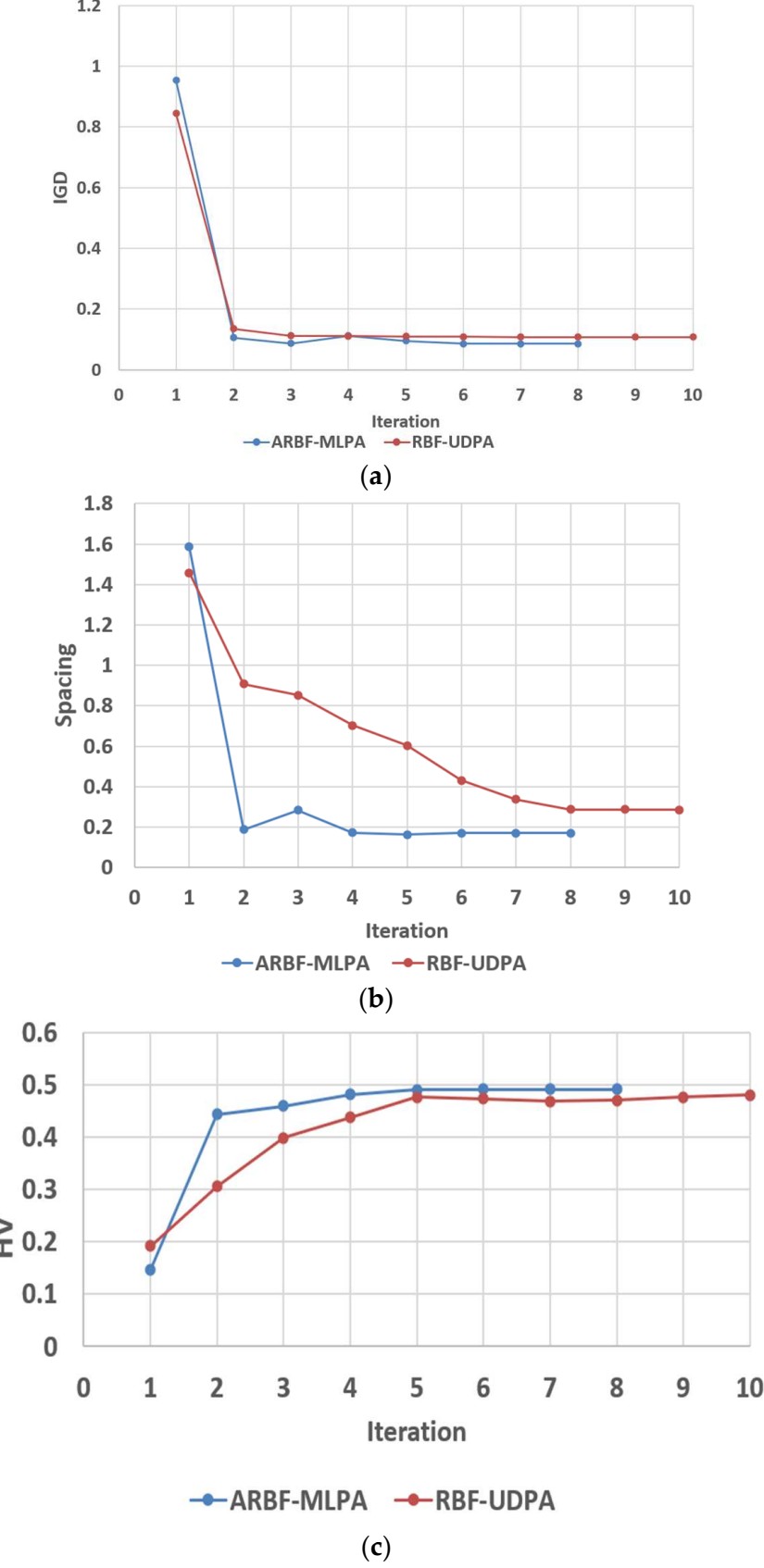

**Figure 5.** (**a**) IGD under function ZDT2. (**b**) Spacing under function ZDT2. (**c**) HV under function ZDT2.

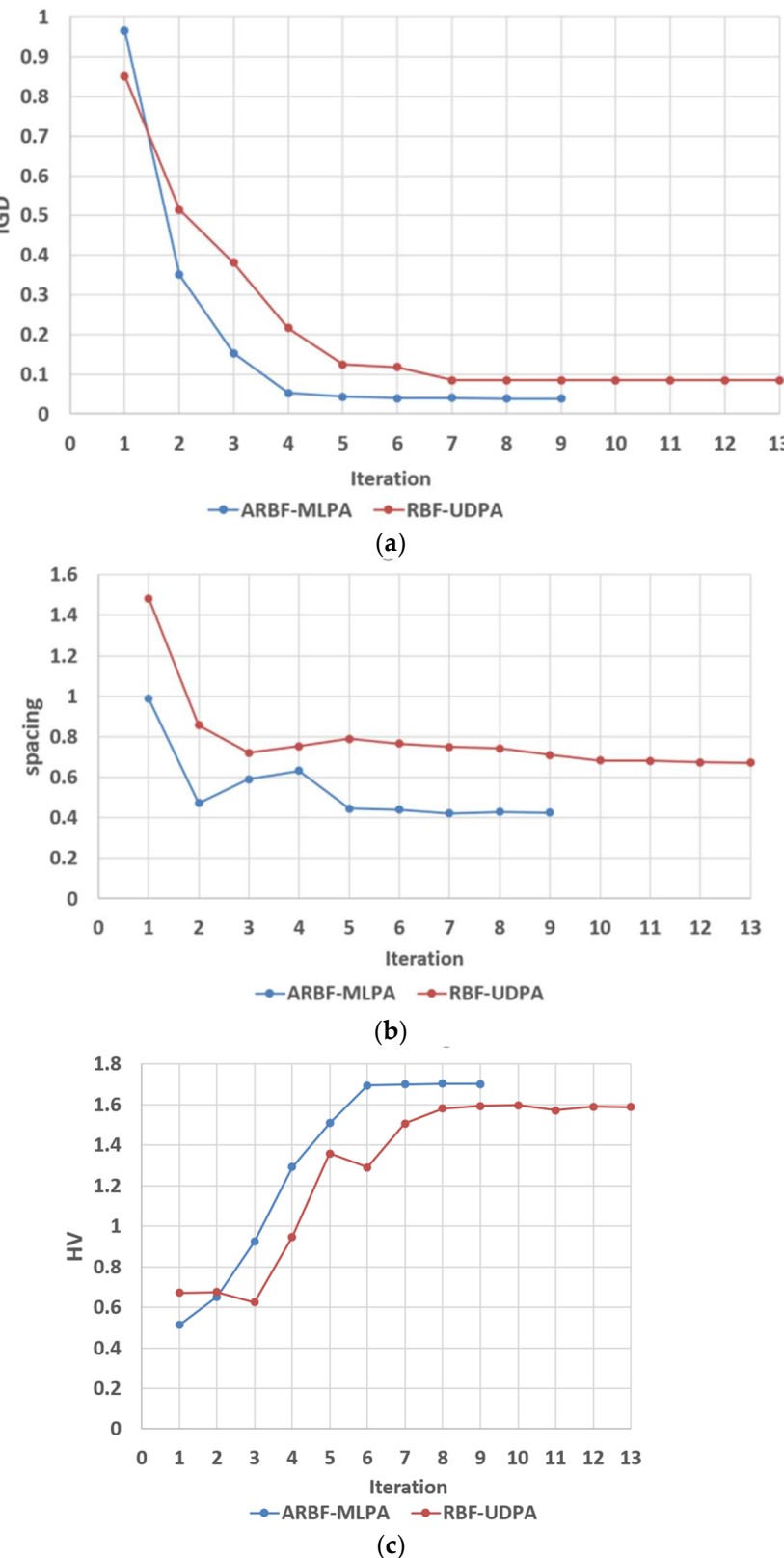

**Figure 6.** (**a**) IGD under function ZDT3. (**b**) Spacing under function ZDT3. (**c**) HV under function ZDT3.

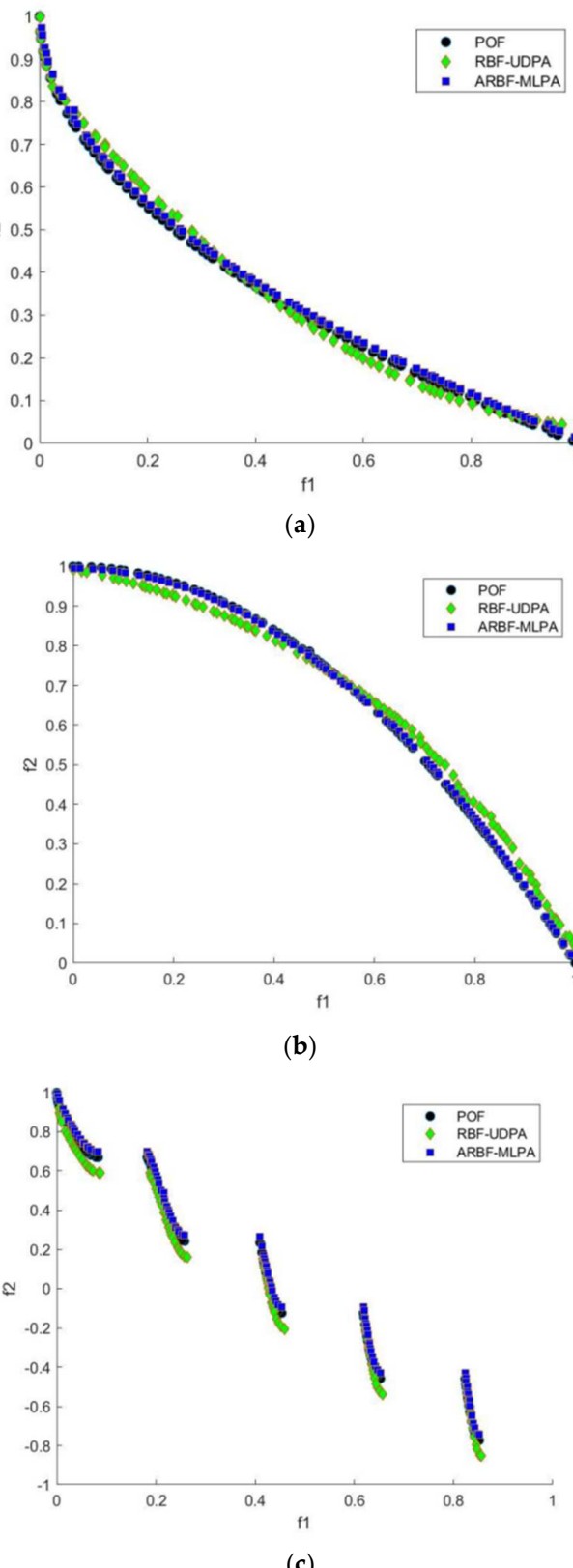

**Figure 7.** (**a**) Front comparison under function ZDT1. (**b**) Front comparison under function ZDT2. (**c**) Front comparison under function ZDT3.

## 3. Selection of Finite Element Model (FEM) and Selection of Optimization Objectives

### 3.1. Establishment of FEM for Vehicle-Side Crash

As a high-precision aggregate, automobiles will suffer a great nonlinear deformation instantaneously when a crash occurs. In this sense, establishing a high-precision whole-vehicle FEM is the precondition to optimization. A definite process and standards have been formulated in order to establish an accurate CAE FEM, see Figure 8.

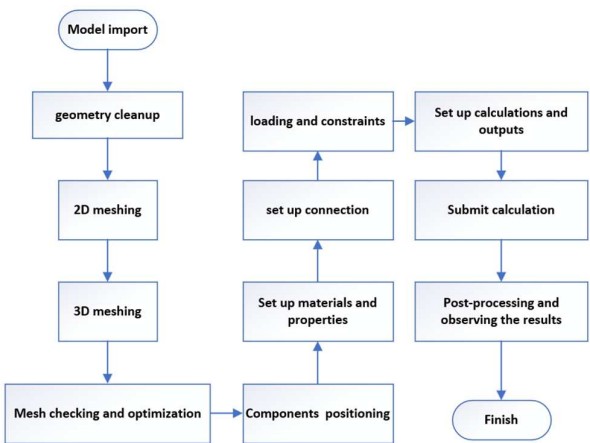

**Figure 8.** The establishment process of whole vehicle FEM.

There are total 886,138 elements in the whole-vehicle model described in this paper, including 42,685 triangular elements, accounting for 4.82% of the total number of elements, which conforms to the empirical value that the number of triangle meshes does not exceed 5% of the total number of mesh elements. Therefore, the simulation calculation of this model is effective, and can be applied in subsequent design and optimization. Figure 9 shows the FEM for automobile safety simulation analysis as described in this paper.

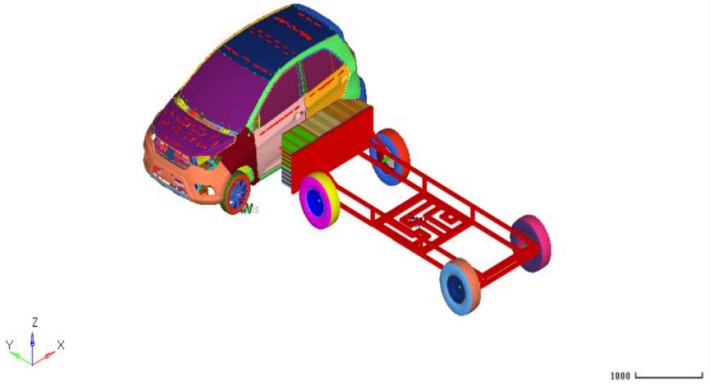

**Figure 9.** FEM of whole-vehicle-side crash.

### 3.2. Validation of FEM

As required by the validation of FEM simulation accuracy, apart from the control of triangle quantity, it is also necessary to ensure energy conservation throughout the crash simulation process. LS-DYNA software applies Single-Point Gauss Integral method in the calculation of FEM, as this method is very suitable for the structural simulation calculation of whole-vehicle crash in which severe deformation occurs. However, this method may produce an hourglass pattern during the calculation.

Hourglass deformation will cause the failure of the simulation result, so we should try to protect against it. However, as hourglass deformation is unavoidable, what we need do is to control it within a certain range (10%). Unless this target value is achieved, the accuracy of the simulation model cannot be ensured. Figure 10 shows the energy change

curve during the crash of the model described in this paper. Hourglass contributes 5.5% of total energy; within the required range, this means that, from the perspective energy, the model is validated.

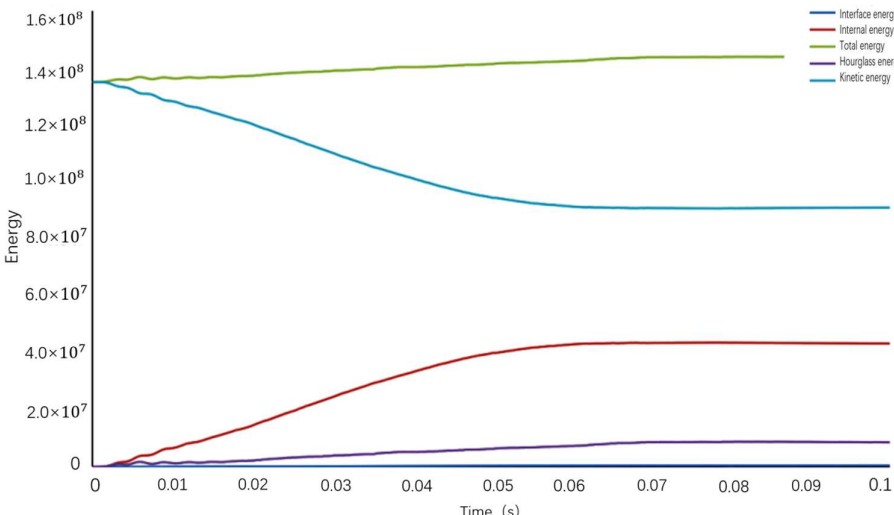

**Figure 10.** Energy curve of vehicle-side crash.

To save computation time, element density is adjusted automatically in simulation by controlling mass scaling. However, the change of element density will increase the mass of the model, therefore affecting computation accuracy. Only by ensuring that the mass increase is within a certain range while reducing computation time can we ensure the effectiveness of the model. According to experience and the literature, we should control the mass increase within 5%. As shown in Figure 11, the mass increase is 195 kg, contributes 3.6% of total, so it meets the requirement.

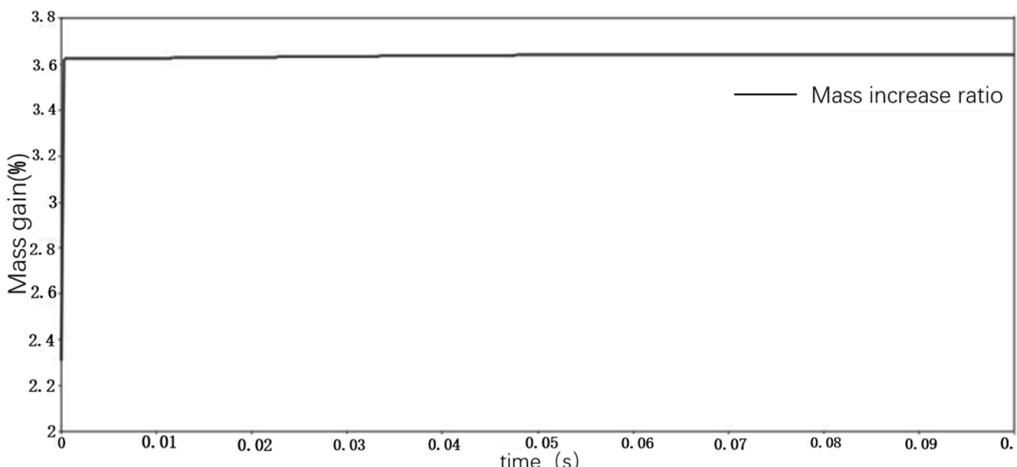

**Figure 11.** Percentage of mass increase.

To sum up, the FEM established under this paper is effective.

### 3.3. Selection of Optimization Objectives

### 3.3.1. The Amount of Chest Intrusion

While vehicle-side impact occurs, the driver in the vehicle is prone to intrusion injury at head, chest, abdomen and pelvis. These parts are mapped to the corresponding positions of B-pillar and measure the intrusion amount at each part indirectly by installing a spring element at B-pillar. Figure 12 shows the mapping between spring element of B-pillar and

the corresponding body parts; Figure 13 shows the deformation and displacement curve of each part during the crash. Among all these parts, the chest is weakest as the external pressure it can withstand is the lowest, so the optimization focuses on the deformation of chest as target. Among all, the amount of chest intrusion during the crash is 161.45 mm at maximum.

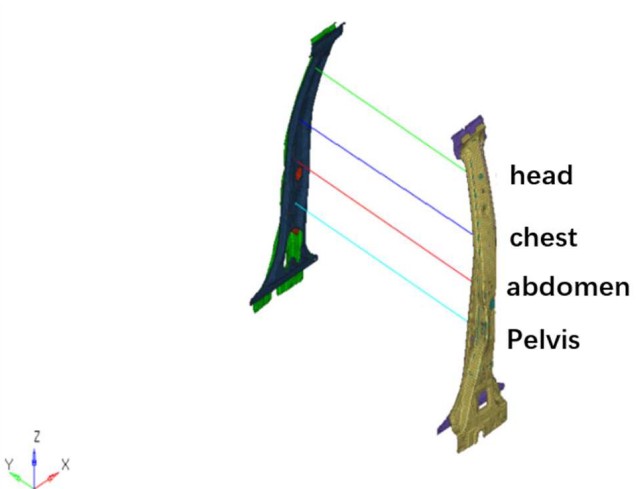

**Figure 12.** Measurement position of spring element of automobile B-pillar.

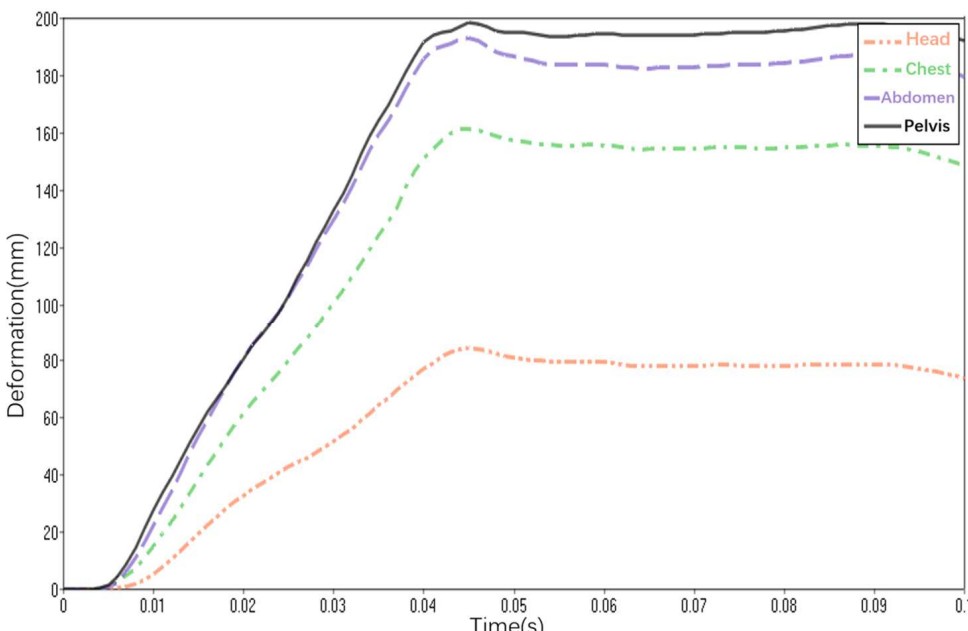

**Figure 13.** Amount of deformation of spring element at B-pillar.

### 3.3.2. Peak Value of Collision Force

During a side impact of a vehicle, the passengers in the car also suffer a very strong collision force. For this reason, control of the peak impact force is also a key objective in protecting the passengers in vehicles. Figure 14 shows the collision force change curve in simulation of vehicle-side impact. It can be seen from the Figure 14 that the peak the collision force is 161.50 KN.

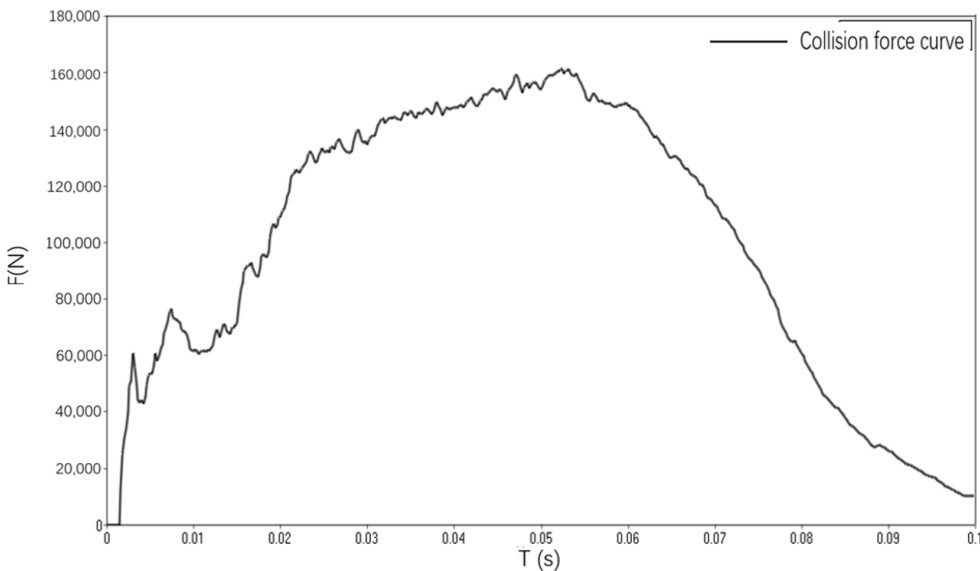

**Figure 14.** Vehicle-side collision force.

To sum up, this paper selects the amount of chest intrusion and the peak collision force as the optimization objectives for whole-vehicle-side crash.

## 4. Optimization Design of Vehicle-Side Crashworthiness

According to the above analysis on vehicle-side crash, too high intrusion in vehicle-side crash is caused by inadequate side rigidity, and automobile B-pillar is a main bearing part for side crash, so the optimization analysis need focus on B-pillar. For this reason, we select the inner and outer boards of B-pillar, B-pillar stiffener, and the upper and lower doorsill beams of the vehicle as the designed variables for side crashworthiness, and take peak collision force and the amount of chest intrusion as the crashworthiness optimization objective.

### 4.1. Selection of Designed Variables

Figure 15 shows the original measurements of the designed variables upper doorsill beam ($t_1$), lower doorsill beam ($t_2$), outer board of B-pillar ($t_3$), inner board of B-pillar ($t_4$) and stiffener of B-pillar ($t_5$): 1.5 mm, 1.2 mm, 1.5 mm, 0.75 mm and 1.5 mm respectively.

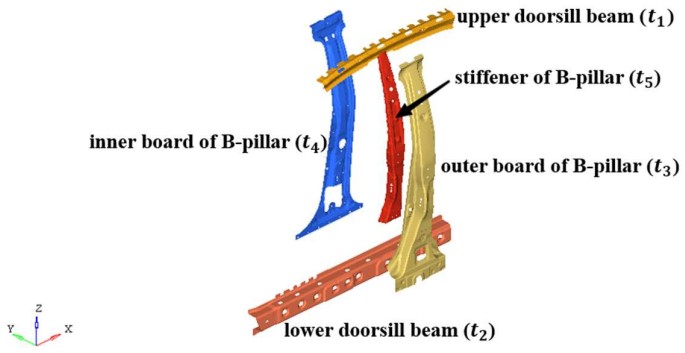

**Figure 15.** Diagram of designed variables for vehicle-side crash.

### 4.2. Establishment of Multi-Objective Model

When a vehicle-side crash occurs, the major impact on the passengers in the vehicle comes from the intrusion at B-pillar and the impacting force during the crash. As a part of the optimization engineering, we need improve crash worthiness while realize the lightweight characteristics of the vehicle. Therefore, we need set the initial weight m*

as constraint upper limit. In order to improve crashworthiness, reduce peak collision force and reduce the amount of chest intrusion, we need increase the design space of designed variables and increase the design space within ±30% of original measurement on the premise of fulfilling manufacturing and design requirements, to achieve optimal crash performance. To sum up, as below is the model of whole-vehicle crash worthiness multi-objective design established under this paper:

$$
\begin{cases}
\text{Min } \{F_{max}(t_1, t_2, t_3, t_4, t_5), -D(t_1, t_2, t_3, t_4, t_5)\} \\
\text{s.t.} \begin{cases}
m(t_1, t_2, t_3, t_4, t_5) - m^* < 0 \\
1.05 \text{ mm} < t_1, t_3, t_4 < 1.95 \text{ mm} \\
0.84 \text{ mm} < t_2, t_4 < 1.56 \text{ mm}
\end{cases}
\end{cases} \tag{4}
$$

wherein, $F_{max}$ represents the peak collision force during the crash, D represents the amount of chest intrusion during the crash, m represents the total mass of designed variables.

### 4.3. Model Solution

First of all, establish an adaptive radial basis neural network based on the 35 sample points obtained using OLHS method and the response value using finite element software, as shown in Table 3. In the process of establishing neural network, randomly select 25% of the sample as test set, 75% as training set.

**Table 3.** 35 sample points and relative response values.

| Number | $t_1$ (mm) | $t_2$ (mm) | $t_3$ (mm) | $t_4$ (mm) | $t_5$ (mm) | $F_{max}$ (N) | D (mm) |
|---|---|---|---|---|---|---|---|
| 1 | 1.66 | 0.84 | 1.66 | 0.83 | 1.32 | 161,732 | 152.144 |
| 2 | 1.45 | 1.03 | 1.82 | 0.62 | 1.10 | 163,497 | 153.255 |
| 3 | 1.82 | 1.35 | 1.29 | 0.56 | 1.61 | 159,388 | 162.529 |
| 4 | 1.74 | 1.18 | 1.34 | 0.97 | 1.58 | 157,333 | 157.281 |
| 5 | 1.61 | 0.93 | 1.71 | 0.87 | 1.87 | 160,340 | 156.951 |
| 6 | 1.10 | 1.07 | 1.58 | 0.94 | 1.74 | 160,169 | 160.868 |
| 7 | 1.29 | 1.01 | 1.26 | 0.68 | 1.08 | 159,750 | 154.788 |
| 8 | 1.95 | 1.29 | 1.74 | 0.76 | 1.84 | 158,812 | 158.481 |
| 9 | 1.87 | 1.05 | 1.39 | 0.61 | 1.16 | 162,148 | 153.432 |
| . . . | . . . | . . . | . . . | . . . | . . . | . . . | . . . |
| 33 | 1.84 | 1.54 | 1.50 | 0.81 | 1.42 | 157,375 | 154.149 |
| 34 | 1.79 | 1.41 | 1.18 | 0.80 | 1.92 | 157,580 | 163.254 |
| 35 | 1.90 | 1.31 | 1.87 | 0.64 | 1.29 | 158,880 | 155.757 |

Find the solution of the above multi-objective optimization model using the ARBF-MLPA method described in this paper. To ensure the result is stable, conduct computation 45 times per generation, and work out the average value as the final solution. Use HV criteria as the standard for evaluating convergence and judge whether HV converges using the expression (5). If yes, stop the computation. As shown in Figures 16 and 17, when the algorithm iteration reaches generation 6, according to HV convergence criteria, the model has met the convergence requirement.

At the moment, total 36 new sample points have been added in the entire iteration process.

$$
\frac{HV_{i+1} - HV_i}{HV_i} \leq \eta \tag{5}
$$

wherein, $HV_{i+1}$ and $HV_i$ represents the HV value of generation i + 1 and i respectively. When the computation results consecutive three generations are lower than the set threshold value $\eta$, it can be concluded that the algorithm has converged. Here $\eta$ is set to be 0.04.

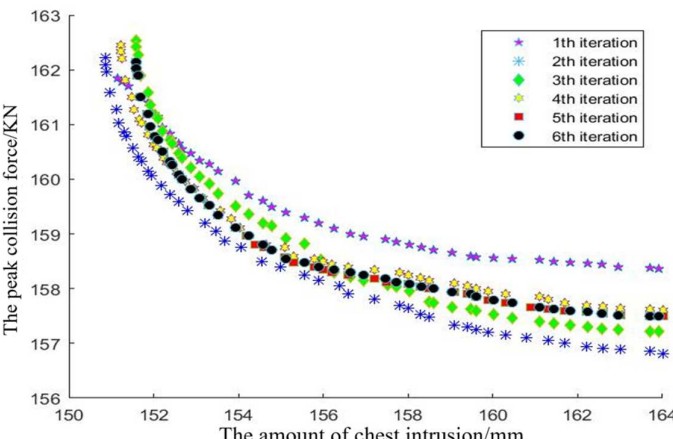

**Figure 16.** The iteration process of Pareto optimal front in finding solution of vehicle-side multi-objective crash worthiness using ARBF-MLPA method.

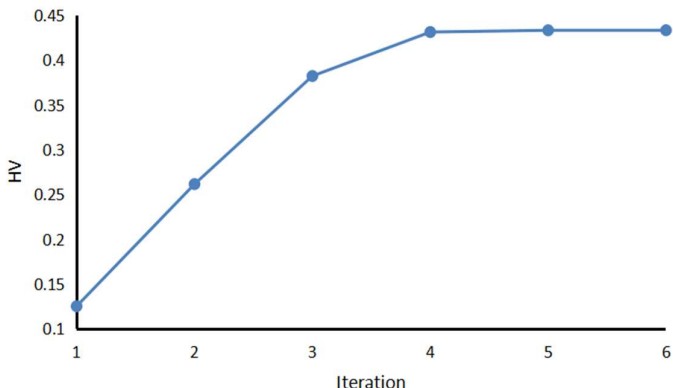

**Figure 17.** The iteration process of HV in finding solution of crash worthiness using ARBF-MLPA method.

### 4.4. Results and Error Analysis

High-precision mathematical agent modelling is the basic premise of optimization calculation. For validation of agent model in this paper, this article selects $R^2$ (determination coefficient) and RMSE (root-mean-square error) precision for the model of the evaluation index. The specific calculation formula is as follows:

$$R^2 = \frac{\sum_{i=1}^{n}(\hat{y}_i - \overline{y})^2}{\sum_{i=1}^{n}(y_i - \overline{y})^2} \tag{6}$$

where $y$ is the value to be fitted; $\overline{y}$ is the mean; $\hat{y}$ is the fit value; $\sum_{i=1}^{n}(\hat{y}_i - \overline{y})^2$ is regression sum of squares (SSR); $\sum_{i=1}^{n}(y_i - \overline{y})^2$ is total sum of squares (SST).

$$RSME = \sqrt{\frac{\sum_{i=1}^{n}(\hat{y} - \overline{y})^2}{n}} \tag{7}$$

In general, the closer $R^2$ is to 1, the smaller the value of RMSE, and the higher the accuracy of the mathematical model.

We carried out mathematical analysis and verification on the accuracy of the model. The following table shows the accuracy verification results of the optimal approximate front when the model converges:

As can be seen from the Table 4, after six iterations of point selection, the approximate model established through ARBF has reached a very high accuracy and can completely replace the finite element model for optimization design.

**Table 4.** The RMSE and $R^2$ of ARBF model at convergence.

| Response Value | RMSE | $R^2$ |
|---|---|---|
| The amount of chest intrusion | 0.1875 | 0.9865 |
| Peak value of collision force | 0.1924 | 0.9737 |

After 6 iterations, the model has converged, and it is now completely possible to select the optimal solution on the basis of this model. Figure 18 shows the optimal Pareto front edge of iteration 6. Researchers should then select the most satisfactory Pareto scheme, namely the Knee point [29], as the final design value of the optimization. The target response values obtained in the optimization design are maximum chest intrusion ($d_{max}$) is 153.92 mm and peak collision force ($F_{max}$) is 159.12 KN.

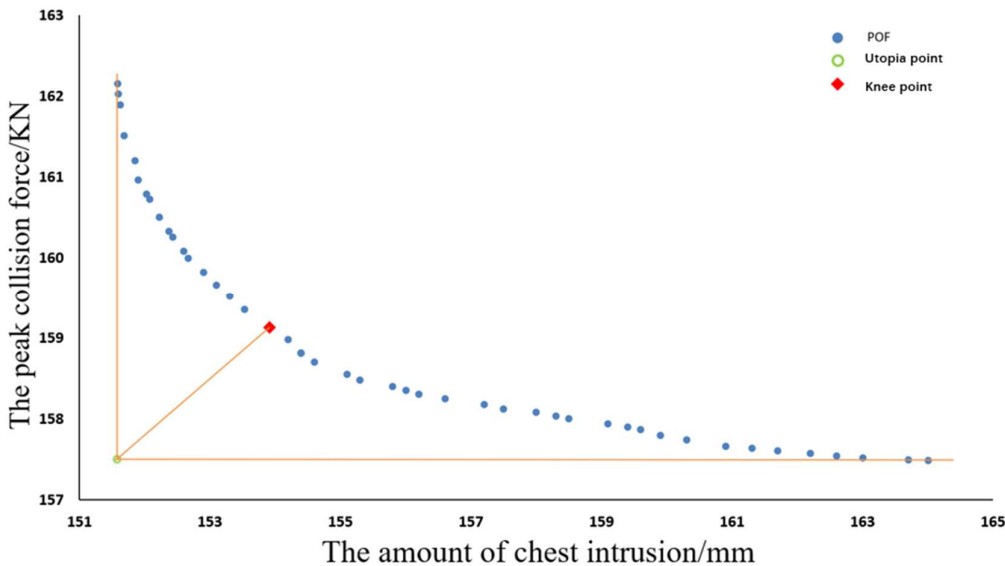

**Figure 18.** Pareto optimal approximate front and Knee point of multi-objective optimization design for vehicle-side crash.

Table 5 shows the comparison between the corresponding response value of optimal design point and the response value before optimization, and the computation results of finite element for the optimization scheme. The optimal design results obtained after optimization are: $t_1$ = 1.76 mm, $t_2$ = 0.93 mm, $t_3$ = 1.12 mm, $t_4$ = 0.86 mm, $t_5$ = 1.20 mm. As shown in Table 5, both the errors of finite element and the errors of ARBF-MLPA are within the acceptable range. After the optimization, in whole-vehicle-side crash, the peak collision force decreased by 2.11%, chest maximum intrusion decreased by 4.32%, while total mass decreased by 14.05%.

According to Figures 17 and 18, the ARBF-MLPA method converges in the sixth generation, and the convergence speed and accuracy are improved compared with the traditional point method. According to Table 5 and the optimization results, it can be seen that the ARBF-MLPA method can be well used in vehicle-side crashworthiness assessments. Compared with the traditional finite element calculation, it can not only greatly reduce the calculation time, but also ensure the accuracy. Therefore, the results prove that the hypothesis proposed in this paper is valid, and the method in this paper can also be generalized and used in other disciplines.

**Table 5.** Comparison of whole-vehicle crash worthiness before and after optimization.

| Parameters | | EFM | ARBF-MLPA | EFM after Optimization | Optimal Ratio |
|---|---|---|---|---|---|
| **Designed variables** | $t_1$ (mm) | 1.5 | 1.76 | 1.76 | |
| | $t_2$ (mm) | 1.2 | 0.93 | 0.93 | |
| | $t_3$ (mm) | 1.5 | 1.12 | 1.12 | |
| | $t_4$ (mm) | 0.75 | 0.86 | 0.86 | |
| | $t_5$ (mm) | 1.5 | 1.20 | 1.20 | |
| **Response value** | $d$ (mm) | 161.45 | 154.48 | 152.12 | −4.32% |
| | Error | \ | 1.53% | | |
| | $F$ (KN) | 161.50 | 158.10 | 152.12 | −2.11% |
| | Error | \ | 3.78% | | |
| | $m$ (Kg) | 11.67 | 10.03 | 10.03 | −14.05% |

*4.5. Computational Cost Analysis*

The computational cost is relatively low for this method.

From the performance test of the algorithm, under the two test functions of ZDT1 and ZDT2, the traditional point-adding method converges in the 10th generation. Meanwhile, the proposed algorithm converges in its 10th iteration and the method in this paper converges in the 9th iteration. Under the ZDT3 test function, the traditional point-adding method converges in the 13th iteration, and the method in this paper converges in the 9th iteration.

For the application of a multi-objective optimization design of vehicle-side crashworthiness, the whole calculation process converges in the 6th iteration, and only 36 points are selected to update the model by using the MLPD in the calculation process.

In conclusion, compared with the traditional point-adding method and the research on vehicle-side crashworthiness, the calculation cost of the proposed method is lower.

**5. Conclusions**

1.  This paper proposed to conduct optimization design for vehicle-side crash safety using the ARBF-MLPA method combined with NSGA-II optimization algorithm; compared to traditional static metamodels, this method improves the solution-finding efficiency significantly.

2.  It analyzed and discussed the point-adding strategy of ARBF-MLPA method, model updating and optimizing process. Then, it validated the effectiveness and fast convergence speed of the algorithm using the mathematical examples of different optimal fronts edge.

3.  It conducted optimization design for vehicle-side crash using ARBF-MLPA, selected peak collision force and the amount of chest intrusion as optimization targets, reducee peak collision force and the amount of chest intrusion while ensuring lightweight of vehicles. After 6 iterations solved the model convergence, results showed that, the model after optimization has higher accuracy than finite element solution and has better crashworthiness too, while also realizing the lightweight characteristic of the vehicles.

4.  The ARBF-MLPA, combined with the NSGA-II optimization method described in this paper, achieved good effects in vehicle-side crashworthiness optimization analysis. In order to verify the universality of this method, it is necessary to apply this method widely to solution-finding methods of various multi-objective problems.

**Author Contributions:** Conceptualization: D.G. and B.Y.; Methodology, D.G.; Software, G.C.; Validation, B.Y., G.C. and Q.L.; Formal Analysis, Q.L.; Investigation, B.Y.; Resources, D.G.; Data Curation, B.Y.; Writing—Original Draft Preparation, G.C.; Writing—Review & Editing, B.Y.; Visualization, L.; Supervision, D.G.; Project Administration, D.G.; Funding Acquisition, D.G. All authors have read and agreed to the published version of the manuscript.

**Funding:** This work is supported by the research fund of National Natural Science Foundation of China (Grant Nos. 52175239) and by the Shanghai Automotive Industry Technology Development Foundation (Grant No.1744).

**Data Availability Statement:** The data used to support the findings of this study are available from the corresponding author upon request.

**Conflicts of Interest:** The authors declare that they have no conflict of interest.

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
