# Peer review of "Multi-Objective Optimization Design of Vehicle Side Crashworthiness Based on Machine Learning Point-Adding Method"

_applsci, doi:10.3390/app122010320_

Round 1

Reviewer 1 Report

After carefully reading this paper, I came up with the following comments: 

A.      This contribution is vague and not clearly articulated. Please create a bullet point list at the end of introduction to highlight the specific contributions of this paper

B.       Please highlight the hypothesis of this work too after listing the contributiosn

C.       For such papers, there should be clear methodology on how descriptive and statistical test are used to ensure the results are statistically sound.

D.      One important measure of numerical methods is their computational cost. There should be clear discussion around this and proper analysis

E.       Please include a paragraph to justify how your comparative analysis will lead to a mature conclusion.

F.       Case studies should be clearly referenced and cited if from literature.

G.      Justification of the quantity and quality of case studies for investigating the hypothesis of this paper should be given too

Reviewer 2 Report

1.    This study proposed a machine learning multi-objective algorithm for the optimization of vehicle side crash-worthiness. This manuscript is well organized and writing, and may be accepted for publication after careful responses and revision on the following comments.  

2.    Highlight the novelty of the manuscript in the introduction section. 

3.    Are design samples only generated in the first population during implementing multi-objective optimization?

4.    Describe the optimization procedure in detail rather than NSGA-II algorithm shown in Figure 2.

5.    What is the purpose to regress an RBF neural network? Is the RBF model regressed in each generation in multi-objective optimization? If the RBF model is first regressed using the generated sample points by OLHS method, is the accuracy of the RBF model sufficient to obtain optimal design of the physical model?

6.    Page 5: it is not clear how to add the high-quality point on optimal front edge obtained through ARBF-MLPA to the model to be built next time.

7.    Which software were the authors used to implement the proposed optimization algorithm? Describe the implementation method in detail.  

Reviewer 3 Report

The authors propose to combine the ARBF-MLPA method with the NSGA-II optimization algorithm to optimize the design for vehicles side crash safety. The results look promising.

The manuscript is well structured and described. However, I suggest the following amendments:

- Review Table 2.2. The functions in the "Function expression" column do not look complete.

- Review the bibliographic references.

- Typos must be corrected, and the authors must avoid repeating the same word several times in the same paragraph or section (e.g. "invented" in the Introduction section).
